# Virulence, Antibiotic Resistance, and Phylogenetic Relationships of *Aeromonas* spp. Carried by Migratory Birds in China

**DOI:** 10.3390/microorganisms11010007

**Published:** 2022-12-20

**Authors:** Bing Liang, Xue Ji, Bowen Jiang, Tingyu Yuan, Chao Lu Men Gerile, Lingwei Zhu, Tiecheng Wang, Yuanguo Li, Jun Liu, Xuejun Guo, Yang Sun

**Affiliations:** 1Changchun Veterinary Research Institute, Chinese Academy of Agricultural Sciences, Changchun 130117, China; 2Key Laboratory of Jilin Province for Zoonosis Prevention and Control, Changchun 130117, China; 3Ruminant Diseases Research Center, College of Life Sciences, Shandong Normal University, Jinan 250013, China; 4Center for Animal Disease Control and Prevention of Yi Jin Huo Luo Banner, Ordos 017299, China

**Keywords:** *Aeromonas* spp., migratory bird, antibiotic resistance, virulence genes, biofilm, phylogenetic tree

## Abstract

This study aimed to evaluate antimicrobial resistance, virulence, and the genetic diversity of *Aeromonas* isolated from migratory birds from Guangxi Province, Guangdong Province, Ningxia Hui Autonomous Region, Jiangxi Province, and Inner Mongolia in China. A total of 810 samples were collected, including fresh feces, cloacal swabs, and throat swabs. The collected samples were processed and subjected to bacteriological examination. The resistance to 21 antibiotics was evaluated. A phylogenetic tree was constructed using concatenated *gltA*-*groL*-*gyrB*-*metG*-*PPSA*-*recA* sequences. Eight putative virulence factors were identified by PCR and sequencing, and a biofilm formation assay was performed using a modified microtiter plate method. In total, 176 *Aeromonas* isolates were isolated including *A. sobria*, *A. hydrophila*, *A. veronii*, and *A. caviae*. All isolates showed variable resistance against all 16 tested antibiotic discs, and only one antibiotic had no reference standard. Six kinds of virulence gene markers were discovered, and the detection rates were 46.0% (*hlyA*), 76.1% (*aerA*), 52.3% (*alt*), 4.5% (*ast*), 54.0% (*fla*), and 64.2% (*lip*). These strains were able to form biofilms with distinct magnitudes; 102 were weakly adherent, 14 were moderately adherent, 60 were non-adherent, and none were strongly adherent. Our results suggest that migratory birds carry highly virulent and multidrug-resistant *Aeromonas* and spread them around the world through migration, which is a potential threat to public health.

## 1. Introduction

Wild animals are one of the main reservoir hosts of bacterial antibiotic resistance genes and play a key role in their transmission [1]. Among wild animals, those of the class Aves occupy a wide range of niches all over the world. Birds, especially waterfowl, have long-distance migration, complex physiological characteristics, unique diets, and lifestyles, and live in diverse natural environments such as remote mountains and lakes [1]. As in all vertebrates, trillions of microorganisms live in birds’ intestines [2]. The most frequently studied bacteria are *Escherichia coli*, *Salmonella enterica*, and *Campylobacter jejuni* [3], and little attention has been paid to *Aeromonas* spp. The harm inflicted by *Aeromonas* on migratory birds cannot be ignored [4]; consequently, a comprehensive study on *Aeromonas* in migratory wild birds is urgently needed.

*Aeromonas* spp. are ubiquitous, Gram-negative rod-shaped and non-spore-forming bacteria approximately 1–3 μm in length, commonly found in freshwater and estuarine environments and isolated from drinking water, groundwater, lakes, rivers, sewage effluents, soil, fish, invertebrates, and raw food [5,6,7]. *Aeromonas* can cause intestinal and extraintestinal diseases, ranging from relatively mild illnesses such as gastroenteritis, urinary tract and wound infection to life-threatening conditions such as septicemia, peritonitis, endocarditis, meningitis, and necrotizing fasciitis [7,8,9,10]. These bacteria are considered to be zoonotic and opportunistic pathogens, which can pose a serious threat to public health and safety.

Some studies have suggested that the virulence genes of *Aeromonas* constitute an important index for the assessment of pathogenicity, and the co-actions of multiple virulence genes can directly or indirectly enhance the pathogenicity [11]. This study mainly detected the distribution of eight related virulence genes, hemolysis A (*hlyA*), aerolysin (*aerA*), cytotoxic enterotoxin (*act*), cytotonic gene (*alt*), enterotoxic activities (*ast*), flagella (*fla*), lipase (*lip*), and type III secretion system genes (*ascF-G*) [8,9,10,11,12,13,14,15,16,17,18,19]. The flagellum is involved in biofilm formation as an adhesion factor. Biofilm can improve the resistance of bacteria to antibiotics and increase the chance of persistent infection.

Antibiotic-resistant pathogens pose a serious threat to human and animal health. Antimicrobial resistance in bacteria is not merely a regional or national problem, but an urgent problem to be solved all over the world [16]. The dissemination and increasing populations of antimicrobial-resistant bacteria have been reported globally, presenting one of the greatest global challenges facing public health [17]. Increasing evidence shows that human multidrug-resistant bacteria can disseminate to the natural environment, maintain clinical resistance, and further spread to wild animals in the environment. Strains of *Aeromonas* have been reported that can undergo interspecies transmission between wildlife and humans [18]. There are many mobile DNA elements in *Aeromonas*, which allow the horizontal transfer of resistance and virulence genes between *Aeromonas* and other bacteria [19]. In recent years, owing to a considerable number of clinical human cases of *Aeromonas* infection [20,21,22], an increasing number of studies have begun to focus on *Aeromonas* spp.

In this study, we aimed to (i) evaluate the levels of virulence and antimicrobial resistance in *Aeromonas* isolates from migratory birds and (ii) their biofilm-forming capabilities, and (iii) construct a neighbor-joining tree of 176 *Aeromonas* strains, with access to epidemiological background data of resistant *Aeromonas* spp. from migratory birds.

## 2. Materials and Methods

### 2.1. Sample Collection

In our study, from January 2018 to June 2019, a total of 810 samples were collected, including fresh feces, cloacal swabs, and throat swabs (Approved by the State Forestry Administration and the Laboratory Animal Welfare and Ethics Committee of the Changchun Veterinary Research Institute, Chinese Academy of Agricultural Sciences (AMMS-11-2020-11)). The sampling locations were Nanning in Guangxi Province, Zhaoqing in Guangdong Province, Suichuan in Jiangxi Province, Tianhu Wetland Park, Yellow River beach wetland and Qingtongxia Nature Reserve in Ningxia Hui Autonomous Region, Ordos City, Honghaizi Wetland Park, and Dali Lake in Inner Mongolia, respectively. The main species of migratory birds included Ciconiiformes, Anseriformes, Gruiformes, and Charadriiformes. All migratory bird samples were collected under the supervision of the Wild Animal Sources and Diseases Inspection Station, National Forestry and Grassland Bureau of China, and did not cause any harm to the animals. The swabs were placed in 20% glycerol physiological saline at –80 °C for a week and transported to the laboratory on dry ice for further processing.

### 2.2. Isolation and Identification of Aeromonas spp. and Antimicrobial Susceptibility Testing

The samples were cultured on a selective *Aeromonas* agar base medium (QingDao Hope Bio, Qingdao, China), and secondary purification was carried out. The strains were subcultured on BHI agar plates and incubated for 16–18 h at 36 ± 1 °C [15]. The NMIC/ID four panel of the BD Phoenix TM-100 automated microbial identification system (Becton Dickinson Company, East Rutherford, NJ, USA) was used to determine the sensitivity of *Aeromonas* to 21 types of antibiotics, shown in Appendix A. The instrument automatically interprets the results, and the determination of the drug sensitivity results is based on the standards of the American Association for Clinical and Laboratory Standards [23,24]. A single colony of *Aeromonas* isolates was cultured in 1 mL BHI broth overnight at 36 ± 1 °C, 600 μL enrichment solution and 400 μL 50% glycerol normal saline were added and the sample and stored at –80 °C. 

### 2.3. Detection of Virulence Determinants by PCR Assay

The suspension of the *Aeromonas* isolates was centrifuged for 2 min at 12,000× *g*; the supernatant was discarded and resuspended in 200 μL sterile deionized water, and incubated in boiling water for 10 min. The tube was again vortexed and centrifuged for 2 min at 12,000× *g* [8]. The supernatant was transferred to a fresh tube and stored at –20 °C and acted as a DNA template for the subsequent detection of virulence genes. Eight virulence genes were detected by PCR. The primers were synthesized by Kumei Bio Inc and their sequences are shown in Table 1. Each PCR amplification of virulence genes was performed in a reaction volume of 25 mL, containing 12.5 μL Taq PCR MasterMix (2×), 0.5 μL 10 μM primer, 1 μL DNA template, and 10.5 μL ddH_2_O. The template was amplified under the following cycling conditions: pre-denaturation at 95 °C for 5 min, 30 cycles of denaturation at 95 °C for 30 s, annealing at 55–66 °C for 30 s, and extension at 72 °C for 1 min, followed by a final cycle at 72 °C for 5 min. *A. hydrophila* ATCC 7966 was included as a positive control and ultrapure deionized water as a negative control in the PCR reactions. The PCR-positive products were further confirmed by sequencing.

### 2.4. Detection of Biofilm Formation

Biofilm formation by the *Aeromonas* isolates was assessed using the microtiter plate method described by Isoken H. Igbinosa, with modifications [25]. Ninety-six-well cell culture plates (Costar, Washington, DC, USA) were filled with 180 μL of BHI and inoculated with 20 μL of *Aeromonas* isolates grown overnight and standardized to 0.5 McFarland. Plates were incubated for 12 h at 37 °C. Positive control wells contained *A. hydrophila* ATCC 7699 and the negative control wells contained uninoculated BHI. Studies were done in triplicate for each well. The contents of each well were discarded and the wells washed three times with sterile phosphate-buffered saline (PBS). After airdrying, the wells were stained with 200 mL of 1% crystal violet for 30 min. The wells were carefully washed with PBS to remove the excess stain. Plates were allowed to dry at room temperature. Dye bound to adherent cells was resolubilized with 200 mL of absolute ethanol. The optical density (OD) value of the biofilm was determined at 450 nm using an Enzyme standard instrument (BioTek Instruments Inc, Winooski, VT, USA) and the average OD of each duplicate result was taken, including positive and negative controls. Biofilm formation was confirmed according to the criteria of Afreenish Hassan et al. [26], as shown in Table 2.

### 2.5. Phylogenetic Analysis

The primer sequences of *gyrB*, *groL*, *gltA*, *metG*, *ppsA*, and *recA* were synthesized [12] and the PCR products were forward sequenced. The consensus sequence for each gene fragment was determined by the ClustalW2 alignment. Multiple alignments containing the concatenated sequences were straightforward and were performed according to the genomic gene order, *gyrB*, *groL*, *gltA*, *metG*, *ppsA*, and *recA*. All analyzed MLST sequences had the same length (2893 nucleotides) [12]. For phylogenetic analysis, concatenated sequences were aligned and analyzed using MEGA X, the phylogenetic tree was constructed using the neighbor-joining method, and iTOL was used for the phylogenetic tree display and annotation. Bootstrap analyses were performed using 1000 replications for NJ.

### 2.6. Statistical Analysis

The dates were analyzed using SPSS version 25.0. A nonparametric one-way analysis of variance (ANOVA) was used to analyze the results. A *p* value of < 0.05 was considered statistically significant, while a *p* value of < 0.01 was considered highly significant.

## 3. Results

### 3.1. Isolation of Aeromonas spp.

It was assumed that large, flat, and green single colonies with dark green centers (2–3 mm in diameter) represented *Aeromonas*. A total of 176 strains of *Aeromonas* were isolated from 810 samples (feces, cloacal swabs, and throat swabs) from migratory birds from five provinces in China with the isolation rate for each region shown in Figure 1 (the map in Figure 1 was obtained from the USGS National Map Viewer). The results of species identification using the BD Phoenix TM-100 automated microbial identification system demonstrated that the most prevalent species in the fecal samples from migratory birds were *A. sobria* 109 (61.9%), *A. hydrophila* 53 (30.1%), *A. veronii* 9 (5.1%), and *A. caviae* 5 (2.8%). As shown in Table 3, the main epidemic strains are strains of *A. sobria* in various regions. The isolation rates of feces, throat swabs, and anal swabs were 38.4%, 24.7%, and 3.2%, respectively. The isolation rates of different types of samples showed a significant difference (*p* < 0.005).

### 3.2. Resistance Phenotypes of the Aeromonas Strains

As shown in Appendix A, the drug resistance spectrum of the *Aeromonas* strains to 16 antibiotics was confirmed; the drug resistance rates for ampicillin and ampicillin/sulbactam were the highest, at 97.7% (172) and 89.8% (158), respectively. The drug resistance rate to cefazolin was 39.2% (69), and those for gentamicin, cefotaxime, piperacillin, colistin, trimethoprim–sulfamethoxazole, chloramphenicol, and tetracycline ranged from 8.0% to 14.8%, while the drug resistance rates for ceftazidime, cefepime, aztreonam, amoxicillin-clavulanate, and ciprofloxacin ranged from 0.6% to 4.0%. The distributions of antimicrobial susceptibility, to the antibiotics tested, of *A. sobria*, *A. hydrophila*, *A. veronii*, and *A. caviae* isolated from migratory bird samples are presented in Figure 2. Each type of *Aeromonas* has significant differences in antibiotic response (*p* < 0.05). *A. hydrophila* and *A. sobria* had the most kinds of antibiotic resistance and the highest drug resistance rate, so that *A. hydrophila* and *A. sobria* had the most serious drug resistance problem. However, the antibiotic resistance of *A. caviae* should not be overlooked. 

Olumide Odeyemi [27] determined the statistics of the drug resistance spectrum of *Aeromonas* spp., and it was observed that isolates were susceptible to amikacin, imipenem, meropenem, and levofloxacin. In this study, the 176 strains showed 31 kinds of drug resistance spectrum. Resistance to more than three types of antibiotic was defined as multi-antibiotic resistance (MAR ≥ 0.25) [28]; as shown in Appendix A, the multidrug resistance rate was 0.23% (41). Among these samples, there were 28 kinds of drug resistance phenotypes in the north, of which A (48.2%) and B (13.4%) were the main phenotypes. The most resistant strains were resistant to eight types of antibiotics. There were seven kinds of drug resistance phenotypes in the south, with A (41.7%) and B (16.7%) as the main phenotypes. The multidrug resistance rate was 25% (3); these strains were resistant to six kinds of antibiotics at most.

### 3.3. Virulence Determinants of Aeromonas spp.

Six kinds of virulence gene markers were discovered in the 176 isolates, and the detection rates were 46.0% (*hlyA*), 76.1% (*aerA*), 52.3% (*alt*), 4.5% (*ast*), 54.0% (*fla*), and 64.2% (*lip*), as shown in Appendix A. The *ascF-G* and *act* genes were not detected in any isolate. A comparison of virulence gene rates in different regions of China showed that the relative abundance of virulence genes was higher mainly in Dali Lake, Ningxia, and Ordos. The detection rates were higher in *A. hydrophila*: *hlyA* 44 (83.0%), *aerA* 42 (79.2%), *alt* 47 (88.7%), *ast* 6 (11.3%), *fla* 43 (81.1%), and *lip* 42 (79.2%). By contrast, in *A. sobria*, *hlyA* was found in 30 (27.5%), *aerA* 82 (75.2%), *alt* 34 (31.2%), *ast* 2 (1.8%), *fla* 39 (35.8%), and *lip* 60 (55.0%). In *A. veronii*, *hlyA* was found in 7 (77.8%), *aerA* 8 (88.9%), *alt* 7 (77.8%), *fla* 8 (88.9%), and *lip* 6 (66.7%). In *A. caviae*, *aerA* was found in 2 (40%), *alt* 5(100%), *fla* 5 (100%), and *lip* 5 (100%).

### 3.4. Biofilm Formation by Aeromonas spp.

In this study, *Aeromonas* isolates were categorized into four groups according to the strength of biofilm formation. The weak producers of biofilm, which make up 58.0% (102) of the isolates, were most common among the groups classified; 8.0% (14) demonstrated moderate capabilities for biofilm production, whereas 34.1% (60) did not form significant biofilm. No strains formed strong biofilms, as shown in Table 4. Among the four species of *Aeromonas*, only 14 *A. sobria* were classified as moderately biofilm-forming; no *A. caviae* strains tested formed biofilm.

### 3.5. Phylogenetic Relationships

The portions of the six housekeeping genes were successfully amplified and sequenced in all 176 strains, except that the *gyrB* and *recA* were not amplified successfully in the *Aeromonas* strains NM31 (*A. sobria*) and NM25Z (*A. sobria*). The phylogeny of the 174 *Aeromonas* strains was analyzed by constructing a neighbor-joining tree from the 2893-bp concatenated sequences (Figure 3). The tree revealed three major phylogroups, of which *A. sobria*, *A. hydrophila,* and *A. caviae* are the main strains. Among the four *Aeromonas* species, *A. sobria* and *A. veronii* clearly showed a high degree of relatedness, and *Aeromonas* isolates from different regions also had close genetic relationships. The phylogenetic tree revealed strong nodal support for three major lineages. All the isolates in different branches were easily distinguishable.

In the phylogenetic trees constructed from individual gene trees, the *PPSA* gene tree has the highest consistency, the *gyrB* gene tree, *groL* gene tree and *gltA* gene tree are generally consistent, while there was a large difference in the *recA* gene tree and *metG* gene tree when compared with the concatenated sequence tree. Most remarkably, the phylogenetic tree constructed for the housekeeping gene seems to be related to that of the virulence genes.

## 4. Discussion

From a public health perspective, the long-distance migration of waterfowl provides a mechanism for the global transmission of bacterial pathogens [29]. The East Asian-Australasian Flyway (EAAF) is one of four globally recognized flyways for migratory waterbirds, and the geographical location of China means that it plays an important role in the migration route of the EAAF migratory birds [30]. In this study, we focused on the differences among *Aeromonas* species of migratory birds in different regions of China. One of the main results of the study was that *Aeromonas* spp. carried by migratory birds are mainly *A. sobria*, *A. hydrophila*, *A. veronii*, and *A. caviae*, in agreement with Cardoso et al. [4]. In addition to *A. sobria*, all *Aeromonas* species identified herein have been related to cases of gastroenteritis in humans, which indicates clinical importance [4]. At the same time, the infection and pathogenicity of *A. sobria* cannot be overlooked, there are increasing studies that have reported clinical cases of *A. sobria* infection [31]. This shows that humans pose a serious threat to migratory birds and pose a serious threat to public health. 

The second main result of the study is that the isolation rate of *Aeromonas* spp. in the north was significantly greater than that in the south. Microbial communities are highly dynamic, affected by both internal (e.g., physiological state, sex, breeding status, genetic predisposition) and external (e.g., season, location, diet, social interactions) factors [32]. There are differences in environmental temperature, air humidity, day length, geography, and environment between the south and the north of China [33]. A possible reason for the discrepancy is that the timing of large-scale dissemination of *Aeromonas* spp. was inconsistent with the time of sampling. Although the climate in the south is more suitable for the growth of *Aeromonas*, the isolation rate of *Aeromonas* in the north was higher in this study. The reason for this result is that the migration of migratory birds to the south occurs in January, when the metabolism and reproductive capacity of *Aeromonas* are low, while the sampling time in the north was in April, when the metabolism and reproductive capacity of *Aeromonas* are increased. Although the sampling time was different in the north and south, the main epidemic strain was *A. sobria*, which is inconsistent with the results of Yongxiao Zhu [33].

The third main result of the study is that the isolation rate from cloacal swabs was lower than that of oropharyngeal swabs and the isolation rate from swimming birds was higher than that from wading birds. This may be because of the ubiquitous presence of *Aeromonas* spp. in surface waters, and the feeding mode of swimming birds. There are many factors affecting the composition and diversity of intestinal bacterial communities of migratory birds, including diet, environment, and season [34]. 

In this study, the *Aeromonas* spp. isolated from wild birds in most areas had some drug resistance. Resistance rates differed slightly between regions: the drug resistance of *Aeromonas* species in Ningxia was the most serious, and the antibiotic resistance trends were in line with the results of Zhang et al. [35]. Normally, *Aeromonas* spp. have inherent resistance to ampicillin, cefazolin, and amoxicillin/clavulanic acid. However, although the resistance rate of *Aeromonas* strains to ampicillin and amoxicillin/clavulanic acid reached more than 90% in this study, some *Aeromonas* strains were susceptible to ampicillin and amoxicillin/clavulanic acid, which is consistent with the results of Yano and Li [9,36]. The rate of drug resistance to cefazolin was low, which is consistent with the results of Wang Shuxian and Zhang Piao [37,38]. Fortunately, the rate of resistance of *Aeromonas* to other antibiotics is also low [1]. We found that the sensitivity of different species of *Aeromonas* spp. to antibiotics may differ, which indicates that we should be able to select drugs according to the species of *Aeromonas* which is causing infection [36]. The drug resistance spectrum of *Aeromonas* spp. might vary according to the geographical area. Five strains of *Aeromonas* (two strains of *A. sobria* were isolated from Qingtongxia Nature Reserve and three strains of *A. hydrophila* were isolated from Yellow River beach wetland) were isolated from the feces of migratory birds in Ningxia. The multidrug-resistant strains were resistant to nine kinds of antibiotics: gentamicin, cefazolin, cefotaxime, ampicillin, piperacillin, ampicillin-sulbactam, trimethoprim-sulfamethoxazole, chloroamphenicol, ciprofloxacin, and tetracycline. Although multidrug-resistant *Aeromonas* spp. have been widely reported [39], the emergence of multidrug resistance in *Aeromonas* spp. carried by migratory birds deserves more attention. Therefore, it is necessary continuously to monitor the emergence of determinants of antibiotic resistance and the drug resistance library [26]. The *Aeromonas* spp. carried by migratory birds mainly come from the environment. Therefore, it is urgent to evaluate the drug resistance of *Aeromonas* spp. in the environment. It is not only necessary to pay attention to the important role of migratory birds in the transmission of drug-resistant bacteria, but also to reduce the use of antibiotics in order fundamentally to reduce the transmission of drug-resistant bacteria. 

The virulence genes were analyzed from the regional aspect: there were significant differences in the content of different virulence genes in the five regions. The relative abundance and diversity of virulence genes were the highest in Dali Lake; Ningxia was second. The comparison of virulence genes between the two regions in Inner Mongolia showed that although the diversity of virulence genes was the same, the differences in relative abundance were significant. It may be that a large number of migratory birds died in the Inner Mongolia region of China between 2017 and 2018; although *Aeromonas* spp. has not been isolated and identified, we cannot rule out that *Aeromonas* is one of the causes of disease in migratory birds [40]. From a comparative species perspective, *A. hydrophila* and *A. veronii* have high relative abundance and diversity of virulence. The number of strains containing three or more virulence genes in *A. hydrophila* and *A. veronii* were higher than in the other two species. This further confirmed that *A. hydrophila* and *A. veronii* are highly pathogenic *Aeromonas* spp. The virulence genes *hlyA*, *aerA*, *alt*, *fla*, and *lip* had very high detection rates. The rate of detection of *ast* was relatively low, and other virulence genes were not detected. This is consistent with the study of Li F, where *hlyA*, *aerA*, *alt*, *fla*, and *lip* were highly prevalent in *A. hydrophila* and *A. veronii*, but some genes such as *laf* and *ascF-G* were not common in all species [35]. It is worth noting that there were 25 virulence-gene combination patterns among the 176 isolates in our study, of which the *aerA*, *aerA*-*lip*, and *hlyA*-*aerA*-*alt*-*fla*-*lip* virulence-gene combination patterns were dominant. For the combination pattern of *aerA* and *aerA*-*lip* virulence genes, *A. sobria* was the main epidemic strain, while *A. hydrophila* was the main epidemic strain with the pattern *hlyA*-*aerA*-*alt*-*fla*-*lip*. Considerable differences in virulence-gene combination patterns may exist between different species. These differences may contribute to diversity in pathogenesis among different *Aeromonas* spp. [41]. Furthermore, the differential expression of virulence genes may be an important factor in the pathogenesis of *Aeromonas* spp. [6]. An experimental study showed a positive correlation between the number of virulence genes and the pathogenicity of *Aeromonas* spp. [42].

The ability of *Aeromonas* spp. to form biofilms could increase their drug resistance and the possibility of persistent infection [43]. The data of this study show that, under the same conditions, the ability of different species to form biofilm can be the same, and the ability of the same species to form biofilm can differ. In the study of Chenia and Duma, the ability of *Aeromonas* strains isolated from freshwater fish and seawater to form organisms is similar to the finding of this study [44]. In our study, 176 *Aeromonas* isolates were not strongly adherent, and most of them were only weakly adherent. These results are similar to the study of Isoken H Igbinosa [25]. Although recent research has shown that the *fla* gene is related to biofilm formation [43], no correlation was found in our study. This indicates that the complexity of biofilm formation of *Aeromonas* cannot be determined simply by the genotype.

Owing to the complexity of the classification of *Aeromonas* spp., many researchers have studied the phylogenetic tree of *Aeromonas* spp. using many different methods; 16S rDNA sequencing is the most common molecular tool in bacteriological classification, but it has proved to have several disadvantages. Housekeeping genes are considered to be suitable markers for phylogenetic analysis [45], and the phylogenetic tree can be more accurately constructed by sequence concatenation of six housekeeping genes [12]. In this study, a concatenated set of housekeeping genes (*gltA*-*groL*-*gyrB*-*metG*-*PPSA*-*recA*) was used to construct a phylogenetic tree by the neighbor-joining method. We evaluated the phylogenetic tree of 174 *Aeromonas* isolates based on the concatenated gene sequences (Figure 3), and revealed that these isolates were closely related and included *A. sobria* (107 isolates), *A. hydrophila* (53 isolates), *A. veronii* (9 isolates), and *A. caviae* (5 isolates). Although four *Aeromonas* species were identified according to the biochemical tests, there were only three large outgroups in the phylogenetic tree of housekeeper gene construction. This is because *A. veronii* is within the *A. hydrophila* cluster in three of the individual gene loci making up the MLST (*gyrB*, *gltA,* and *metG*) [46]. *A. veronii* is divided into *A. veronii* bv. *Sobria* and *A. veronii* bv. *veronii*, and some studies have shown that *A. veronii* bv. *Sobria* and *A. sobria* belong to the same taxon [12]. The genetic relationship between *A. caviae* and the other three strains is relatively distant. However, the use of several gene loci may mask the evolutionary history of individual genes, although an increase in the number of gene loci increases the resolution of the analysis by joining the combined capacities of all molecular clocks [46]. Between the phylogenetic tree of the *PPSA* and the concatenated phylogenetic tree, there are few differences in the overall clustering of most strains. This may be because the *PPSA* gene locus is the least variable. Some studies have shown similar results with the *recA* and the *metG* gene loci, which is consistent with this study (Appendix A) [46]. This study verified that *PPSA* can be used to construct a more accurate phylogenetic tree. The results showed that the genetic distances in the phylogenetic tree constructed from housekeeping genes are related to the number of virulence genes (Figure 3, Appendix A). It was found that the phylogenetic tree constructed from housekeeping genes for *A. sobria* and *A. hydrophila* seems to be related to virulence genes. The relationship between *A. sobria* carrying multi-virulence and major *A. hydrophila* phylogroups was closer; *A. hydrophila* with few or no virulence genes is more closely related to major *A. sobria* phylogroups.

## 5. Conclusions

We obtained 176 *Aeromonas* isolates from migratory bird samples, and high genetic diversity was observed in these isolates. Virulence genes were examined by PCR, indicating that *Aeromonas* spp. Are well-equipped with potential virulence genes including *hlyA*, *aerA*, *alt*, *ast*, *fla*, and *lip*, and pose a risk to human health. When measuring antibiotic resistance to nine distinct antibiotic classes, 94.3% of the strains were found to be MAR (≥3). Our findings demonstrate that migratory birds can be reservoirs of virulent and multidrug-resistant *Aeromonas*, and act as a vehicle for the transfer of different genotypes of *Aeromonas* and antibiotic-resistant determinants to regions worldwide through migration. Strengthening the directed surveillance of wild migratory birds can effectively anticipate disease outbreaks, allowing for preventive or mitigating measures to be taken.

## Figures and Tables

**Figure 1 microorganisms-11-00007-f001:**
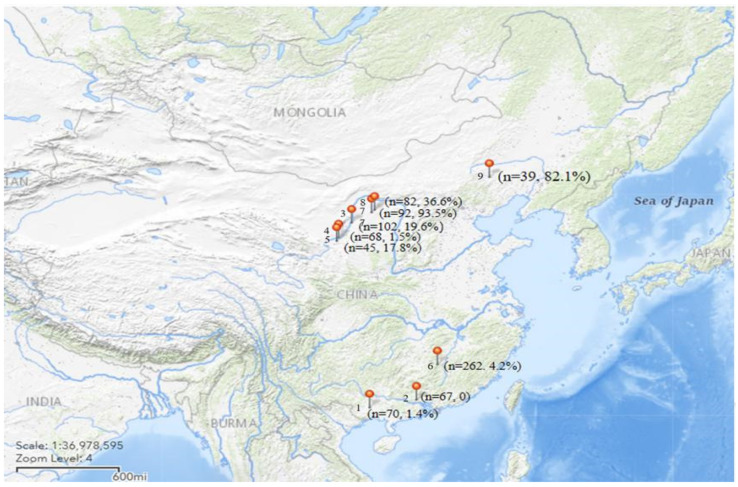
The isolation rate of *Aeromonas* in various regions. The stars indicate different sampling locations and the numbers on the left are sorted by sampling time. Sampling locations: 1. Nanning, Guangxi Province (22.78° N, 108.27° E). 2. Zhaoqing, Guangdong Province (23.10° N, 111.59° E). 3. Tianhu Wetland Park, Ningxia Hui Autonomous Region (37.49° N, 105.68° E). 4. Yellow River beach wetland, Ningxia Hui Autonomous Region (38.83° N, 106.66° E). 5. Qingtongxia Nature Reserve, Ningxia Hui Autonomous Region (37.52° N, 105.13° E). 6. Suichuan, Jiangxi Province (26.32° N, 114.50° E). 7. Ordos City, Inner Mongolia (39.82° N, 109.96° E). 8. Honghaizi Wetland Park, Inner Mongolia (39.55° N, 109.81° E). 9. Dali Lake, Inner Mongolia (43.28° N, 116.74° E).

**Figure 2 microorganisms-11-00007-f002:**
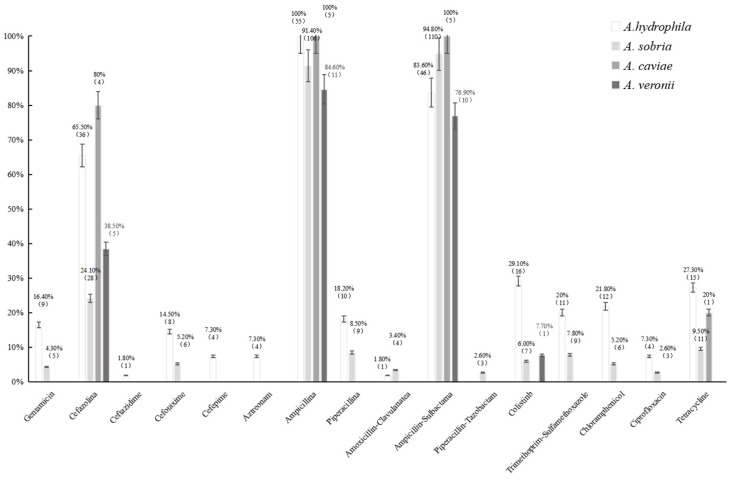
The distributions of antimicrobial susceptibility, to the antibiotics tested, of *A. sobria*, *A. hydrophila*, *A. veronii*, and *A. caviae* isolated from migratory bird samples are presented. Univariate ANOVA revealed that the difference between groups was highly significant for each antibiotic. Other pairwise comparisons within each group were significant.

**Figure 3 microorganisms-11-00007-f003:**
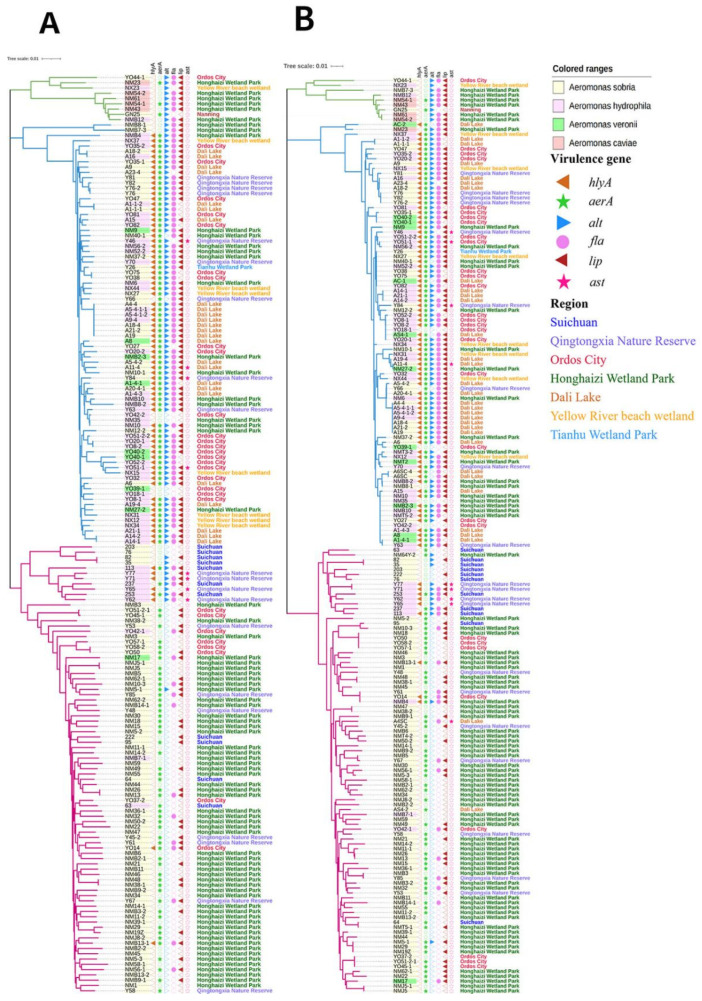
Unrooted phylogenetic trees based on the combined stretch of *gyrB*-*groL*-*gltA*-*metG*-*ppsA*-*recA* (**A**) and *PPSA* (**B**) gene sequences, showing relationships in the genus *Aeromonas* from migratory birds in this study. A solid symbol shows that the strain contains virulence-associated genes; a hollow symbol shows that the strain did not contain the virulence-associated gene. Colored rectangles represent the species of *Aeromonas*. For each strain, the shapes of different colors indicate the presence of the virulence factor genes analyzed in this study. The different color characters represent Aeromonas strains isolated from the seven different regions in China.

**Table 1 microorganisms-11-00007-t001:** Primer sequences of Aeromonas virulence genes.

Target Gene	Function	Nucleotide Sequence (5′–3′)	Product Size (bp)	Reference
*hlyA*	Hemolysis A	F:GCCGAGCGCCCAGAAGGTGAGTT	130	[15]
		R:GAGCGGCTGGATGCGGTTGT		
*aerA*	Aerolysin	F:GC(A/T)GA(A/G)CCC(A/G)TCTATCC(A/T) G	252	[14]
		R:TTTCTCCGGTAACAGGATTG		
*act*	Cytotoxic enterotoxin	F:AGAAGGTGACCACCAAGAACA	232	[14]
		R:AACTGACATCGGCCTTGAACTC		
*alt*	Heat-labile cytotonic enterotoxin	F:TGACCCAGTCCTGGCACGGC	442	[14]
		R:GGTGATCGATCACCACCAGC		
*ast*	Heat-stable cytotonic enterotoxin	F:TCTCCATGCTTCCCTTCCACT	331	[14]
		R:GTGTAGGGATTGAAGAAGCCG		
*fla*	Flagellin	F:TCCAACCGTYTGACCTC	608	[14]
		R:GMYTGGTTGCGRATGGT		
*lip*	Lipase	F:ATCTTCTCCGACTGGTTCGG	382	[14]
		R:CCGTGCCAGGACTGGGTCTT		
*ascF-G*	Type III secretion system	F:ATGAGGTCATCTGCTCGCGC	789	[14]
		R:GGAGCACAACCATGGCTGAT		

**Table 2 microorganisms-11-00007-t002:** Interpretation of biofilm production.

Average OD Value	Biofilm Production
OD ≤ OD_C_	non-adherent
OD_C_ < OD ≤ 2 × OD_C_	weakly adherent
2 × OD_C_ < OD ≤ 4 × OD_C_	moderately adherent
4 × OD_C_ < OD	strongly adherent

Optical density cut-off value (ODc) = average OD of negative control + 3× standard deviation (SD) of negative control.

**Table 3 microorganisms-11-00007-t003:** Sample collection and Aeromonas isolation.

Source						The Sample Type	The Relationship with Human Habitation ^a^	Number of Samples (N)	Species of Aeromonas (*n*)
Sampling Locations			Sampling Time	Taxonomy of Migratory Birds	Ecological Groups of Birds				
The south	Guangdong	Zhaoqing	January 2018	Ciconiiformes	wader	faeces	B	67	—
	Guangxi	Nanning	January 2018	Ciconiiformes	wader	faeces	C	70	A.sobria (1)
	Jiangxi	Suichuan	September 2018	Ciconiiformes	wader	cloacal swabs	C	229	A.sobria (7); A.hydrophila (4)
				unclassified	—			33	—
The north	Inner Mongolia	Honghaizi Wetland Park	April 2019	Ciconiiformes	wader	faeces	B	10	A.sobria (7); A.hydrophila (2); A.veron (1)
				Anseriformes	Natatores			29	A.sobria (19); A.veron (2); A.caviae (1)
				Gruiformes	wader			32	A.sobria (21); A.hydrophila (3); A.caviae (4)
				Charadriiformes	Natatores			14	A.sobria (14); A.hydrophila (5); A.veron (1)
		Ordos City	April 2019	Charadriiformes	Natatores	faeces	B	81	A.sobria (13); A.hydrophila (14); A.veron (3)
		Dali Lake	April 2019	Charadriiformes	Natatores	faeces	A	30	A.sobria (12); A.hydrophila (11); A.veron (2)
	Ningxia Hui Autonomous Region	Tianhu Wetland Park	April 2018	Anseriformes	Natatores	throat swabs and cloacal swabs	A	24 and 24	A.sobria (1 and 0)
				Gruiformes	wader			10 and 10	—
		Qingtongxia Nature Reserve		Anseriformes	Natatores	throat swabs and cloacal swabs	A	46 and 46	A.sobria (13 and 0); A.hydrophila (7 and 0)
				Gruiformes	wader			5 and 5	—
		Yellow River beach wetland		unclassified	—	faeces	B	45	A.sobria (1); A.hydrophila (7)

^a^ The relationship between sampling sites and human habitation. A: Distant; B: Adjacent; C: Within.

**Table 4 microorganisms-11-00007-t004:** Biofilm production of Aeromonas isolates in microtiter plates.

Biofilm Production	*A. hydrophila* (*n* = 53)	*A. sobria* (*n* = 109)	*A. veronii* (*n* = 9)	*A. caviae* (*n* = 5)	Total (*n* = 176)
non-adherent	30.2 ± 1.51%	33.0 ± 1.65%	33.3 ± 1.67%	100 ± 0.05%	34.1 ± 1.71%
weakly adherent	69.8 ± 3.49%	54.1 ± 2.71%	66.7 ± 3.34%	0%	58.0 ± 2.9%
moderately adherent	0%	12.8% ±0.64%	0%	0%	8.0 ± 0.4%

## Data Availability

The data that support the findings of this study are available from the corresponding author upon reasonable request.

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
