# Peer review of "Virulence, Antibiotic Resistance, and Phylogenetic Relationships of Aeromonas spp. Carried by Migratory Birds in China"

_microorganisms, 2022, doi:10.3390/microorganisms11010007_

Round 1
Reviewer 1 Report
The migratory life patterns of migratory birds have become particularly sensitive in the current global epidemic of COVID-19, and it is of great significance to detect the antibacterial, virulence and genetic diversity of migratory birds-associated Aeromonas. The authors performed drug resistance and virulence gene analysis on more than 810 collected samples, and obtained some meaningful results based on phylogenetic analysis of multiple conserved loci.
Some bugs are listed below but not limited to
1. Please indicate the number of repetitions of the phylogenetic tree constructed by MEGA X using the neighbor-joining method.
2. What was the basis for the selection of the sampling sites listed in Section 3.1? Please point out.
3. Relevant statistics, error values ​​for figure 2, table 3, please point out.
4. In the information of cited literature, the name of the journal is sometimes full, sometimes abbreviated, please use a unified format.
5. On the first page of the supplementary materials, please add the title of the article, the author, the author's affiliation and related information.
Reviewer 2 Report
The manuscript addressed the virulence, antibiotic resistance and phylogenetic relationships among Aeromonas spp isolated from migratory birds.
The Manuscript is generally good. However some minor points need to fixed before acceptance. Details of such comments are in the revised file attached.

Reviewer 3 Report
General Comments
Phylogenetic characteristics, virulence and antibiotic resistant 2 of Aeromonas spp. carried by migratory birds in China
Bing Liang, Xue Ji, Bo-wen Jiang, Tingyu Yuan, Chao Lu Men Gerile, Lingwei Zhu, Tiecheng Wang, Yuanguo Li, Jun Liu, Xuejun Guo and Yang Sun
The authors present a study on Aeromonas strains isolated from migratory birds in nine regions of China. Their objective is to determine the extent of resistance to 21 antibiotics and the distribution of six putative virulence genes among the isolates. These data would address the issue of a possible reservoir of potentially pathogenic Aeromonas in avian populations. Overall, the paper is well written, and the data are logically presented. It is of course descriptive biology with good information regarding one aspect of the ecological distribution of Aeromonas.
There are several aspects of the manuscript that require modification before publication. The authors isolate 176 strains of Aeromonas from 810 samples. To determine the taxonomy of the isolates the authors employ the NMIC/ID 4 panel of the BD Phoenix TM-100 automated IS system (this is poorly described in M&M) as well as an MLST sequencing approach targeting six housekeeping genes. The sequenced genes were concatenated and used to cast a phylogenetic tree with MEGA. In surveying for virulence genes, the authors use PCR amplification with previously developed primers followed by a sequencing step to confirm. Given the authors experience with phylogenetic analysis, why aren’t these sequences analyzed with MEGA much the same as the housekeeping genes. No information regarding bootstrapping is presented in the M&M or in the trees although in L362 the authors mention 100% bootstrapping. A more fully described bootstrapping methods and results would be convincing to the reader. There are also several typos and poor sentence constructions that require attention. After these minor modification the manuscript will be ready for publication.
Specific comments
L82. be specific on the time
L111. Why is there no phylogenetic analysis of the virulence genes?
L116 & 117. please check units. Is it ml or µl
L125. Why are you using OD here instead of Absorbance? This will be confusing to most who use the crystal violet staining protocol. If you are reading at a wavelength at which crystal violet has no absorbance, why stain the prep?
L140. No information here about bootstrapping yet you cite the tool in discussions.
L140. In the discussion you note the differences between the trees cast from each of the housekeeping genes. Iapplaud this level of analysis. This does help with your analysis. However, there is at least one important piece ofinformation missing for a critical analysis and that is the number of informative characters for each tree that youcast. Of course, as the number of informative characters diminishes, so to does the robustness of the analysis. Thismay be why you see some gene trees wander away from the composite tree. I also think that presenting the trees isa good strategy. However, you might consider highlighting the substantial differences between the trees in someway. Probably something other than putting more information on the tree. Some measure of congruency,something to help the reader focus on sections of the tree that are wandering depending on the housekeeping geneused.
L147. Are you indicating here that 21% of your samples carried Aeromonas?
L265-267. I find this sentence counterproductive. In fact, the exact opposite of this sentence has moreconvincing data, that humans pose a serious threat to migratory birds.
L359-361. This sentence is not clear. Based on housekeeper genes there are 4 species, the sameapparently as the biochemical tests.
L362. There is no information in M&M on bootstrapping. How many reiterations were calculated?Were all trees bootstrapped?? that is the composite housekeeping gene as well as each geneindividually?
Supplemental Table 2 & 3 need proofing.
Several formatting issues in Sup Table 4
L390-394. Authors contributions need to be rewritten. Several grammatical errors within.
Reviewer 4 Report
Very interesting manuscript for the sharp examination of the entity of the phenomenon of the diffusion of antimicrobial resistance through migratory flows: as authors say, "the emergence of multidrug resistance in Aeromonas spp. carried by migratory birds deserves more attention” and as such is destined to have an ever-increasing interest in a one-health perspective.
The experimental part is well described, with appropriate and up-to-date bibliographic references, however the text dealing with the antimicrobial resistance itself is negatively balanced with respect to the genetic study and deserves more attention
Some observations:
45-47: peritonitis, endocarditis, meningitis are not “relatively mild illnesses”, it is more convenient to add them to “life-threatening conditions”
89-91: please, indicate the tested antibiotics (which are 21 as declared in lines 89-91, or 13 as reported in line 168 in the results and supplementary tab. 2, or 16 as illustrated in fig. 2?) The text is not very clear about it
115: please, check for citations
147-148: could the results be distinguished between throat swabs, cloacal swabs and fresh feces samples, perhaps with some statistical tests? As to fecal samples, what is the meaning of “fresh”? how can the authors be sure of no environmental contamination? And it is possible for the authors to add data on different sub-orders or families, to better characterize and correlate the presence of antimicrobial resistance with habitats and lifestyle of the sampled migratory birds?
168-170: This is a repetition of the Materials and Methods section
192: Can data on which classes of antibiotics show resistance in MAR isolate strains be added?
255-258: perhaps you would like to move this sentence to the Introduction section, to better characterize the interest in Aeromonas and clarify the purpose of the trial (for example, the choice to investigate only aquatic birds or those associated with aquatic environments, as many other avian orders have migrating species)
Notes to supplementary tab. 2: “Reference standards were not used for drug resistance of Aeromonas to mupirocin”: the results of the mupirocin sensitivity test are not present in the supplementary tab. 2, nor in fig. 2, nor in the text: what is the meaning of the sentence?
Reviewer 5 Report
The authors attempted to isolate Aeromonas spp. from migratory birds in the hypothesis that the spread of pathogenic or multidrug-resistant Aeromonas spp. to humans. It is epidemiologically very important to understand the dynamics of pathogenic Aeromonas spp., focusing on migratory birds, which move large distances.
In Materials and Methods 2.4 (p. 4, Table 2), the 'ODc' needs to be explained.
In Results 3.1 (p. 4, L152-153), the number of isolates of each species from 'feces' only is given, but additional Table 1 lists isolates from other materials as well. Please correct the text.
In Results 3.2 (p. 5, L171), ampicillin resistance is 97.7%, which is not something novel, as it is almost 100% for Aeromonas spp., which is pathogenic to humans. Therefore, it is not appropriate to list ampicillin resistance as one of the multidrug-resistant agents; it is better to list resistance to three or more drugs excluding ampicillin resistance.
ref: doi:10.11599/germs.2016.1094
In Results 3.3 (p. 6, L201-212), there is no mention of the act gene at all; please mention it in both Results and Discussion.
In Results 3.5 (p. 6, L227), authors state that they were divided into three groups, but they may be different species of Aeromonas spp..
ref: doi: 10.1111/jam.13887
As mentioned by the authors in Discussion (p. 8, L279-280), Aeromonas spp. are bacteria that is ubiquitous in the environment, especially in freshwater. Not all bacteria isolated at a sample collection site from migratory birds that have eaten or drank may have been transported from other locations. Information on Aeromonas strains already present at the sampling site is also needed. Comparison of both strains of Aeromonas will be important data.
As for the pathogenicity of the isolates to humans, it is difficult to conclude that the isolates are a public health problem without actual confirmation of hemolytic or cytotoxic potential, since many strains carrying pathogenicity-related genes are also isolated from environmental sources.
The potential threat to public health as described in the last line of Abstruct (p. 1, L25) cannot be concluded from this manuscript. It should be limited to the possibility of widespread regional migration of Aeromonas spp. by migratory birds.
Round 2
Reviewer 5 Report
In Results 3.3 (p. 6, L214-225), there is no mention of the act gene at all; please mention it in both Results and Discussion.
In Results 3.4(P. 7, L226-235), in which category of biofilm formation did the positive control A. hydrophila 7699 fall? Is A. hydrophila 7699 a suitable control for biofilm formation? If it is appropriate, is it necessary to compare the data with those of the isolates?
In Results 3.5 (p. 7, L246), authors state that they were divided into three groups, but they may be different species of Aeromonas spp..
ref: doi: 10.1111/jam.13887
As mentioned by the authors in Discussion (p. 9, L298-299), Aeromonas spp. are bacteria that is ubiquitous in the environment, especially in freshwater. Not all bacteria isolated at a sample collection site from migratory birds that have eaten or drank may have been transported from other locations. Information on Aeromonas strains already present at the sampling site is also needed. Comparison of both strains of Aeromonas will be important data.
As for the pathogenicity of the isolates to humans, it is difficult to conclude that the isolates are a public health problem without actual confirmation of hemolytic or cytotoxic potential, since many strains carrying pathogenicity-related genes are also isolated from environmental sources.
The potential threat to public health as described in the last line of Abstruct (p. 1, L27) cannot be concluded from this manuscript. It should be limited to the possibility of widespread regional migration of Aeromonas spp. by migratory birds.
